# Convolutional Neural Network (CNN) Model for the Classification of Varieties of Date Palm Fruits (*Phoenix dactylifera* L.)

**DOI:** 10.3390/s24020558

**Published:** 2024-01-16

**Authors:** Piotr Rybacki, Janetta Niemann, Samir Derouiche, Sara Chetehouna, Islam Boulaares, Nili Mohammed Seghir, Jean Diatta, Andrzej Osuch

**Affiliations:** 1Department of Agronomy, Poznań University of Life Sciences, Dojazd 11, 60-632 Poznań, Poland; 2Department of Genetics and Plant Breeding, Poznań University of Life Sciences, Dojazd 11, 60-632 Poznań, Poland; janetta.niemann@up.poznan.pl; 3Department of Cellular and Molecular Biology, Faculty of Natural Sciences and Life, University of El Oued, El Oued 39000, Algeria; samir-derouiche@univ-eloued.dz (S.D.); boulaares-islam@univ-eloued.dz (I.B.); 4Laboratory of Biodiversity and Application of Biotechnology in the Agricultural Field, Faculty of Natural Sciences and Life, University of El Oued, El Oued 39000, Algeria; nili-m@univ-eloued.dz; 5Department of Microbiology and Biochemistry, Faculty of Sciences, Mohamed Boudiaf-M’sila University, M’sila 28000, Algeria; sara.chetehouna@univ-msila.dz; 6Department of Agricultural Sciences, University of El Oued, El Oued 39000, Algeria; 7Department of Agricultural Chemistry and Environmental Biogeochemistry, Poznań University of Life Sciences, Ul. Wojska Polskiego 71F, 60-625 Poznań, Poland; jean.diatta@up.poznan.pl; 8Department of Biosystems Engineering, Poznań University of Life Sciences, Wojska Polskiego 50, 60-637 Poznań, Poland; andrzej.osuch@up.poznan.pl

**Keywords:** date fruits, Python, artificial intelligence, machine learning, CNN

## Abstract

The popularity and demand for high-quality date palm fruits (*Phoenix dactylifera* L.) have been growing, and their quality largely depends on the type of handling, storage, and processing methods. The current methods of geometric evaluation and classification of date palm fruits are characterised by high labour intensity and are usually performed mechanically, which may cause additional damage and reduce the quality and value of the product. Therefore, non-contact methods are being sought based on image analysis, with digital solutions controlling the evaluation and classification processes. The main objective of this paper is to develop an automatic classification model for varieties of date palm fruits using a convolutional neural network (CNN) based on two fundamental criteria, i.e., colour difference and evaluation of geometric parameters of dates. A CNN with a fixed architecture was built, marked as DateNET, consisting of a system of five alternating Conv2D, MaxPooling2D, and Dropout classes. The validation accuracy of the model presented in this study depended on the selection of classification criteria. It was 85.24% for fruit colour-based classification and 87.62% for the geometric parameters only; however, it increased considerably to 93.41% when both the colour and geometry of dates were considered.

## 1. Introduction

The date palm (*Phoenix dactylifera* L.) is the oldest perennial fruit tree, cultivated since ancient times, originating from Mesopotamia, and characterised by a strongly developed root system that can extend to 1500 m in well-drained and sandy soil. The date palm belongs to the palm family (Arecaceae) and includes over 2500 varieties, with its fruits and seeds (pits) playing an important role in the national economies of many countries, especially the Middle East, South Asia, North Africa, and Central America [1,2,3,4,5,6,7]. Fruits and seeds are a rich and inexpensive source of many nutrients and macro- and microelements. The dominant ingredients of dates are carbohydrates, including soluble sugars and fibre. There is 8.1% to 12.7% of fibre in fruits and about 15% in seeds, of which 84% to 94% is in the form of insoluble fibre. Date fibre has important functional properties, namely the ability to retain water and oil. Date palm fruits are also very rich in phenolic antioxidants (1–2%), mainly flavonoid glycosides such as quercetin, luteolin, apigenin, chrysoeriol, kaempferol, and isorhamnetin [8,9,10,11]. They also contain important nutraceuticals with a wide range of effects, including anti-mutagenic, antioxidant, antimicrobial, anti-inflammatory, hepatoprotective, anticancer, and immunostimulatory properties [12,13,14].

In addition to direct consumption, date fruits are used as a flavouring ingredient in dairy products, sweets, desserts, food mousses, and others. Dried date fruits have a longer shelf life, but their nutritional value is typically lower than that of fresh ones. The quality of dates is usually determined based on their physical properties, such as colour, shape, size, and texture, while their nutritional value is assessed on the basis of their chemical properties and sensory characteristics, which depend primarily on the fruit’s ripeness stage and its variety [9,15]. The physical characteristics of date palm fruits are used as criteria for initial evaluation and qualitative and varietal classification, with colour being one of the more important features.

The nutritional and phytochemical properties of dates vary depending on the harvesting period, variety, and storage and processing method. Date palm fruits develop and ripen in several stages. The first stage is Hababouk, during which the fruits attain their proper shape but are still inedible. The next stage is Kimri, where fruits turn green but are still unripe and inedible. The subsequent stage is Khalal, in which the fruits become full-sized and are already edible, followed by Rutab, where fruits become soft, ranging in colour from brown to black, and are edible. The last stage of date palm fruit development and ripening is Tamr, during which they obtain their colour, brown or black depending on the variety, and their moisture content decreases while sugar content increases [5,16].

Generally, date palm fruits are sorted and classified manually, which is very labour-intensive and time-consuming. Numerous studies are currently being conducted to develop intelligent methods for the accurate, fast, and fully automatic sorting of date palm fruits. Artificial intelligence and machine learning methods can undoubtedly be helpful here.

The dynamic development of artificial intelligence (AI), machine learning (ML), and computer image analysis (CIA) has facilitated the process of extracting quality features of agricultural products, fruits, and vegetables based on shape [17,18,19], colour [20,21], texture [22], and light spectrum [23].

A modern and intensively developing artificial intelligence tool are convolutional neural networks (CNN), characterised by the ability to extract high-level features of objects, thanks to which they are used to solve various multi-level complex problems, such as road traffic monitoring [24], facial and human movement recognition [25], and object recognition [26]. CNN models also play an important role in medicine for understanding the genetic basis and treatment of diseases such as brain [27], breast [28], and skin cancer [29], as well as aneurysms and autism in humans [30,31,32]. CNN is also used in robotics for visual navigation [33], controlling the driving path of autonomous vehicles [34], planning the movement paths of ground robots [35], programming production manipulators [36], and assessing the quality of products, agricultural produce, and other biological materials [37,38]. Modern digital techniques and computer data analysis methods allow for precise and automated control of food quality [39] and the identification of weeds, diseases, or pests in crop plants [40]. Computer image analysis has become one of the main techniques used in agriculture to evaluate seeds and grains in terms of quality losses, quantifying their ripeness stage, and the degree of disease infestation, mechanical damage, or contamination with other plant species. Thanks to the non-invasiveness of these methods and the increasing computing power of computers, image analysis and CNN have a significant advantage over the labour-intensive and expensive methods currently in use [41,42,43]. Artificial intelligence and computer techniques enable the use of precision agriculture technologies in fertilisation, the application of plant protection products, and precise agricultural and technical operations such as sowing, planting potatoes or vegetable seedlings, etc. [44,45,46,47,48,49].

Digital techniques and methods provide new knowledge that can be applied to control the quality of food and agricultural products with high accuracy [50,51,52].

The ongoing research on date palm fruits can be divided into three groups, i.e., classification of varieties [2], analysis of their degree of ripeness [53], and qualitative classification [54]. However, most research on date palm fruit focuses on the qualitative classification of fruits, where the basic criteria are: moisture, sugar content, hardness, surface damage, or disease infection. For example, Ismail and Al-Gaadi [55] classified date palm fruits into moist, semi-moist, and dry using a developed electronic sensor to measure their moisture content. Manickavasagan et al. [56] used hardness as a criterion for classifying date palm fruits and applied linear discriminant analysis (LDA) to distinguish soft, semi-hard, and hard dates. The LDA used by researchers and the related Fisher’s linear discriminant (FLD) allowed for the use/application of machine learning to find a linear combination of features determining the fruit hardness.

Many researchers have proposed methods for classifying date palm fruits according to their surface defects. For example, AI-Janobi [57,58] proposed techniques using grey-level co-occurrence matrices (GLCMs) and computer image analysis for date grading. The GLCMs used by the author were used for the examination of textures, taking into account the spatial relationships of pixels. GLCM functions characterise the texture of an image by calculating how often pairs of pixels with specific values and in a specified spatial relationship appear in the image, creating GLCMs, and then extracting statistical measures from the matrix. Nasiri et al. [54] used a backpropagation neural network to classify dates into three quality classes, considering their size, shape, colour, and surface damage, while Alavi [59] applied fuzzy inference to measure the quality of dates based on features such as length and freshness. Schmilovitch et al. [60] used spectrometry and near-infrared to assess the ripeness of dates, and Zhang et al. [61] classified the ripeness of dates of one variety through colour analysis and backpropagation. Pourdarbani et al. [51] also classified the ripeness of date palm fruit of one variety using taxonomy with colour and texture analysis, including contrast, entropy, and uniformity of dates. A neural network-based date classification system was proposed by Hobani et al. [62] and Albarrak et al. [63]. Haidar et al. [64] used size, shape, colour, and texture to classify images of single fruits into seven classes depending on their variety. Similar to his predecessors, Muhammad [65] also used shape, size, and texture criteria to distinguish four classes of dates based on images of individual dates.

## 2. Materials and Methods

### 2.1. Data Set Preparation

The main objective of the study is to develop a model for the automatic classification of date palm fruit varieties based on their colour and an algorithm for evaluating the geometric parameters of the fruit. The geometric shape of dates determined in the model is their length and diameter. The empirical material in the development of the CNN model and the code automating the classification process consisted of photos of five varieties of date palm fruit, which were taken during the Tamr development phase. The analysed dates were sourced from the south-eastern Algeria region.

The following date varieties were examined: Deglet Nour (DN), Degla Baida (DB), Ghars (GH), Tenicin (TE), and Tantboucht (TA) (Figure 1), and their detailed characteristics are presented in Table 1. The analysed varieties differed in fruit colour, of which the Degla Baida variety was the lightest and the Tenicin variety was the darkest. The fruits of individual varieties also differed in shape, from elongated in the case of Deglet Nour to nearly spherical for Tantboucht, respectively. The highest average length (47.20 ± 0.11 mm) and average weight (9.92 ± 0.44 g) were observed in the Degla Baida variety. Tantboucht had the shortest fruits, with an average length of 25.80 ± 0.08 mm, while Tenicin had the lightest fruits, with an average fruit weight of 7.45 ± 0.79 g.

The smoothness of the fruit surface is extremely important when analysing the geometric shape of date palm fruits and classifying their varieties using machine learning and CNN methods. In the Tamr development stage, the fruits of Tantboucht and Ghars were the most and the least smooth, respectively.

The images of dates were taken with a Xiaomi digital camera equipped with a sensor with a resolution of 12,000 × 9000 (108 million) pixels. Date palm fruits were photographed in a laboratory on a white background and in natural light. Initially, the image files were stored in the camera’s internal memory and then saved in jpg format at a resolution of 216 dpi and dimensions of 1600 × 1204 pixels in the computer’s memory. Each photo was marked with a variety symbol and a reference number, i.e., Deglet Nour: DN001–DN100, Degla Baida: DB001–DB100, Ghars: GH001–GH100, Tenicin: TE001–TE100, and Tantboucht: TA001–TA100.

### 2.2. Loading and Pre-Processing a Data Set

In the first stage of the analysis, the content of the files was checked, and a list of names of the date palm fruit photos was generated using the pathlib library. Then they were visualised, and their size was determined according to code 3 in //github.com/piotrrybacki/date-palm-CNN.git (accessed on 27 September 2023). The images of date palm fruits were loaded into NumPy tables as 8-bit fixed-point numbers with values in the range of <0–255>. The tf.io and tf.image modules from the TensorFlow 2.0 library were used to prepare the dataset. The tf.io module was used to load and store the images of date palm fruits, and the tf.image module was used to decode the raw content and to resize the images.

The displayed list of files shows that the set of input data contain 500 photos of date palm fruits, 100 for each variety, and its size is 188 MB (Figure 2). The photos of the date palm fruits were randomly divided into three subsets, i.e., a training set containing 300 photos (60 photos of each variety) and validation and test sets containing 100 photos each (20 photos of each variety). Listing 1 (//github.com/piotrrybacki/date-palm-CNN.git) shows code 1 for automatic copying of images from the source directory to the training, validation, and test directories.

### 2.3. Image Pre-Processing

Image analysis by a computer system requires conversion from analogue to digital form through digitization. This is achieved through discretization and quantization. Image discretization can be spatial or amplitude-based and is done through two-dimensional sampling at precisely defined points, usually grid nodes. In this paper, spatial discretization was applied to a square grid using code 2 (://github.com/piotrrybacki/date-palm-CNN.git). The results of discretizing the images of date palm fruits are presented in Figure 3.

Quantization, on the other hand, involves dividing the continuous range of colour brightness values into intervals and assigning each point a selected discrete value representing that interval. To evaluate the results of colour quantization (Table 2), the most common metrics are: *MSE* (mean square error) or *RMSE* (root mean square error), and the peak signal to noise ratio (*PSNR*) expressed logarithmically in dB, according to Equations (1)–(3):(1)$$RMSE=\sqrt{\frac{1}{3mn}\sum_{j=1}^{M}\sum_{i=1}^{N}\left[{\left(R_{ij}-R_{ij}^{*}\right)}^{2}+{\left(G_{ij}-G_{ij}^{*}\right)}^{2}+{\left(B_{ij}-B_{ij}^{*}\right)}^{2}\right]},$$
(2)$$MSE=\frac{1}{3mn}\sum_{j=1}^{M}\sum_{i=1}^{N}\left[{\left(R_{ij}-R_{ij}^{*}\right)}^{2}+{\left(G_{ij}-G_{ij}^{*}\right)}^{2}+{\left(B_{ij}-B_{ij}^{*}\right)}^{2}\right],$$
(3)$$PSNR=20\;\log_{10}\frac{255}{RMSE},$$
where: 

*R_ij_*, *G_ij_*, *B_ij_*—the colour components of the original image,

$R_{ij}^{*}$, $G_{ij}^{*}$, $B_{ij}^{*}$—the colour components of the image resulting from quantization,

*M*, *N*—spatial resolution of the image.

**Table 2 sensors-24-00558-t002:** Average values of the discretization process parameters of date palm fruit images.

Image Codes Average	DN001–DN100	DB001–DB100	GH001–GH100	TE001–TE100	TA001–TA100
*PSNR* [dB]	150.55	189.98	235.09	145.87	120.54
*MSE*	89.34	93.67	113.56	85.06	79.57

Figure 4 shows quantized images of dates of five varieties according to algorithm 3 (://github.com/piotrrybacki/date-palm-CNN.git). Each image point was assigned a discrete value.

### 2.4. Defining the Criteria for Classifying Date Palm Fruits

In practice, images are characterised by a large excess of information, so it is extremely important to extract the key features of the imaged objects, in this case, date palm fruits. These features are both criteria for classification and qualitative evaluation.

The basis for defining the criteria for classifying date palm fruits using the CNN model was their colour, size, shape, and surface. Evaluation of the results of discretization and quantization of date palm fruit colour was carried out using the *CIELAB* methodology, based on the theory of Euclidean distance in a perceptually uniform space. The non-uniformity of this space is evident from the fact that perceptual colour differences cannot be clearly determined. Colour can be described mathematically by calculating the parameter Δ*E* according to equation 4 using three components. *L*—brightness (luminance), *a*—colour from green to magenta, *b*—colour from blue to yellow.
(4)$$\Delta E=\frac{1}{MN}\sum_{j=1}^{M}\sum_{i=1}^{N}\left[{\left(L_{ij}-L_{ij}^{*}\right)}^{2}+{\left(a_{ij}-a_{ij}^{*}\right)}^{2}+{\left(b_{ij}-b_{ij}^{*}\right)}^{2}\right],$$
where: 

*L_ij_*, *a_ij_*, *b_ij_*—colour components of the original image,

$L_{ij}^{*}$, $a_{ij}^{*}$, $b_{ij}^{*}$—colour components of the image resulting from quantization.

In the analysis, a criterion was used according to which the absolute colour differences in date palm fruits (Δ*E*) between 0 and 1 are imperceptible and the colour deviation of dates is unnoticeable, characteristic of the same variety, for example. A difference in the range of 1 to 2 indicates a minor deviation, 2 to 3.5 a moderate deviation, 3.5 to 5 a noticeable deviation, while Δ*E* above 5 is a major deviation in colour, which can clearly indicate different varieties of date palm fruits.

The diameter and length of the fruits were adopted as the second criterion for geometric classification. Numerical limits for each analysed variety were determined based on empirical measurements and the average values of these parameters, taking into account measurement errors. Detailed data can be found in Table 3.

The hypothesis was made that incorporating these two classification criteria for date palm fruits would significantly improve the accuracy of the models and streamline the process.

### 2.5. Multilayer Architecture of CNN

The proposed CNN architecture for automatic classification of date palm fruits, marked as DateNET, consisted of an alternating system of five layers: MaxPooling2D, Dropout, and Conv2D with the ReLu activation function, implemented using the Keras interface. MaxPool2D is a class that creates pooling layers, where the parameter strides = 1 is used to configure these layers. The argument pool size = 2 specifies the size of the window used for computing the maximum value. The Dropout class allows the user to define a dropout layer for regularisation, where the argument rate determines the probability of input units being dropped during network learning. When calling this dropout layer, its operation is adjusted by the training argument, which determines if the call is to occur during learning or inference. By default, the Conv2D class assumes that the input data are compatible with the NWHC format, where N stands for the number of images in the group, W and H designate the width and height of the image, and C is the number of channels. As shown in Figure 5, each convolutional layer was followed by a pooling layer for subsampling, i.e., reducing the size of the feature map. Listing 4 posted at //github.com/piotrrybacki/date-palm-CNN.git. shows the code programming this model.

The developed input code automatically transforms the data tensor into feature maps with dimensions of 200 × 200, which ultimately results in obtaining object maps of 7 × 7 just before the flattening layer. This transformation of the input tensor also allowed for increasing the depth of object maps in the network from 32 to 128. The use of binary classification in the developed model allowed the network to be completed with dense layers, i.e., a layer with a dimension of 512 and the ReLu activation function, as well as a layer with a dimension of 1 and the Sigmoid activation function.

The algorithm proposed in the paper automatically sorts dates based on colour and geometric parameters, such as diameter and length, considering the name of each date’s image. It then copies them to the test directory, from which they are retrieved by the DateNET algorithm. Based on the evaluation of the accuracy of date palm fruit classification, loss curves, analysis, and prediction accuracy values are plotted in the next stage according to code 5 (//github.com/piotrrybacki/date-palm-CNN.git).

The proposed DateNET architecture and the algorithm describing it allow for displaying the prediction result as probabilities of date palm fruit images belonging to particular classes (varieties). The tf.argmax function can search for an image with the highest probability of belonging and assign an appropriate label, which is the name of the date variety and the assumed geometric parameters, i.e., the surface of the fruit in the image, its circumference, diameter, and length. This was done for a group of 50 examples, and both input data and predicted labels were visualised according to code 6 (//github.com/piotrrybacki/date-palm-CNN.git).

The performance of the developed date palm fruit classification model was also evaluated using the measures of speed and prediction accuracy. The speed of the model was measured by the classification rate, which expressed the number of assigned images per second and the average classification time for a single date image. The model’s accuracy was evaluated using positive predictive value (PPV), true positive rate (TPR), the result correction factor (f), and its accuracy (ACC). These measures were determined using Equations (5)–(8).
(5)$${PPV}_{X}=\frac{{TP}_{X}}{{TP}_{X}+{FP}_{X}}\;,$$
(6)$${TPR}_{X}=\frac{{TP}_{X}}{{TP}_{X}+{FN}_{X}}\;,$$
(7)$${fscore}_{X}=\frac{1}{\frac{\propto }{{PPV}_{X}}+\frac{\propto }{{TPR}_{X}}}\;,$$
where: 

*TP_x_*—true positive,

*FP_x_*—false positive,

*FN_x_*—false negative,

∝ = 0.5 gives equal weight to TPR and PPV,
(8)$$ACC=\frac{\sum_{i=1}^{n}\frac{{TP}_{i}}{I_{i}}}{n}\;,$$
where: n = no. of classes, I_i_ = no. of images in classe i

For class X, in this analysis of date palm fruit varieties, TPX is a true positive, i.e., the number of images correctly ecognized and assigned to class X. PPVX is the number of true positive results divided by the total number of images predicted to belong to class X. TPRX is defined as the number of true positive results divided by the actual number of images in class X. The f-scoreX is used to combine PPVX and TPRX into a single measure using the harmonic mean. The overall accuracy in Equation (8) was calculated using balanced accuracy, which normalises the true positive result for each class by the number of images in the class and divides their sum by the number of date palm fruit varieties. Balanced accuracy ensures that all classes contribute equally to the overall accuracy calculation, even if the number of fruit images in the classes is unequal. To illustrate the classification accuracy for date palm fruits, confusion matrices of the models were used according to the format in Figure 6.

Additionally, the proposed DataNet model was compared with the ResNet, MobileNet, and ShuffleNet algorithms. A Residual Neural Network (ResNet, ang. Residual Network) was developed by He et al. [66] in order to improve the quality of object classification computer accuracy. ResNet is a deep learning model in which the weight layers learn residual functions with reference to the layer inputs. A residual network is a network with skip connections that perform identity mappings, merged with the layer outputs by addition. ResNet is a multi-variant, multi-layer neural network model that works in the same manner. In our experiment, the ResNet50 model was applied, which was based on a 50 layer neutral network. MobileNet is an advanced convolutional neural network for mobile vision applications, which, according to Srinivasu et al. [67], can be used especially in medical applications, i.e., dentistry. It uses depthwise convolutions to significantly reduce the number of parameters compared to other networks, resulting in a lightweight deep neural network. The third of the analysed models is ShuffleNet, which is a convolutional neural network designed especially for mobile devices with very limited computing power. The architecture of those models utilises two operations, i.e., pointwise group convolution and channel shuffle, to reduce computation costs while maintaining accuracy [68]. 

## 3. Results

The final result of the analysis is the proposal of a DateNET architecture and an algorithm enabling automatic classification of date palm fruit varieties and evaluation of their geometric shape. Table 4 presents a comparison of the changes in the sizes of maps for the developed DateNET model depending on the layer number for 30 epochs. The data show that each hidden layer of the DateNET model results in a reduction of maps, producing a total output of 4,459,332 parameters.

The designed DateNET architecture and the accompanying code allowed for the automatic sorting of images into training, validation, and test directories. Then, the computational algorithm randomly selected the order of unsupervised model training and validation based on the colour and geometric parameters of the fruit. The accuracy of date palm fruit classification for individual varieties depended on the type of criteria chosen. As shown in Figure 7 and Figure 8, the accuracy of the model’s training and validation processes increased with the number of repetitions. However, multiple tests of the model revealed that a limit to the increase in accuracy and decrease in training loss was 30 computational repetition cycles (30 epochs).

Displaying the prediction results as probabilities of belonging to specific classes, according to code 5, was the final result of the analysis. The developed algorithm searched for the image with the highest probability of belonging, gave its value, and assigned the appropriate label, which was the code of the date palm fruit variety. When classifying based on geometric parameters, the algorithm assigned an appropriate label, which was a set of information calculated based on the shape of the image and the number of pixels with the size and shape of the fruit. The label contains information about the surface area of the date, its perimeter, diameter, and length. It also includes an eccentricity index, which is the ratio of the length of the radii from the centre of gravity to the analysed cross-section. An eccentricity index close to 1.00 indicated that the axis of symmetry of the date palm fruit at each *dx* is a straight line.

If there were different results in the two-stage classification, which is based on colour and geometry, the affiliation was determined by the higher probability of the correct result. As presented in Figure 9, the DN image by colour was incorrectly assigned (77.50%), but it was correctly assigned as the DB variety by geometric parameters, with a probability of 81.09%.

Table 5 lists the geometric parameters generated by the proposed model and the constructed DateNET architecture, which were compared with empirical measurements of the diameters and lengths of date palm fruits. Columns 8 and 10 of Table 5 show deviations from the actual average value of the diameter and length of dates. The minus sign before it indicates a smaller diameter, while the plus sign means a diameter larger than the average value. The analyses show that 89.12% of the read values are within the criteria adopted and presented in Table 5. Column 12 also shows that the incorrect image assignment occurred at a low value of the ΔE index, which indicates that the model does not recognise differences in fruit colours and classifies them randomly, making an error. It can also be noticed that the varieties marked as TE (Tenicin) and TA (Tantboucht) were characterised by a high value of the ΔE index.

Figure 10 presents the performance of the proposed DateNET model on the validation data set using the confusion matrix. It is clear that when using the colour of dates for classification, the Deglet Nour variety produced the following coefficients: TPR at 79.00% and PPV at 84.95%. By additionally including geometric parameters in the classification, the TPR index and the PPV index increased to 91.00% and 93.81%, respectively. Even higher values were recorded for the Tantboucht variety, where the TPR and PPV indices were 92.00% and 93.88%, respectively, and with the geometric parameters of date palm fruit, they both reached the value of 96.00%. This demonstrates that out of 100 date images, four were misclassified.

The average time for classifying date images using GPUs depended on the adopted criteria. Images were classified the fastest when based solely on colour (20.43 ms/image), while it took slightly longer, 23.57 ms/image, to assign a photo based on the geometric parameters of the fruit. Photos were classified for the longest time, i.e., 27.88 ms/image, when using a combination of criteria, i.e., colour and geometric parameters (Table 6). However, the combination of classification criteria significantly increased the process accuracy, reaching 93.40% for the proposed CNN model and architecture.

According to Table 7, comparing the DateNet model proposed in this work with randomly selected models, i.e., ResNet, MobileNet, and ShuffleNet, it can be concluded that it is characterised by similar positive predictive value (PPV), true positive rate (TPR), as well as the result correction factor (f) and its accuracy (ACC). However, the main advantage of the proposed DateNet model is that it enables the assessment of geometric date parameters. This allows not only variety classification but also distinction in terms of fruit size. This increases the average classification time compared to the fastest model by 7.12 ms/image.

## 4. Discussion

Comparing the effectiveness of the proposed classification method in this study, which achieved an accuracy of 93.41%, with methods presented in other scientific papers on recognising varieties of date palm fruits and classifying their ripeness stages, identifying palms, and recognising plant sex, it can be concluded that it does not significantly deviate in terms of accuracy and precision. In their studies, Bin Naeem et al. [69] applied supervised machine learning methods (KNN, SVM, Naive Byes, AdaBoost) to identify the sex of date palms. The results obtained by the authors indicated that the SVM algorithm is the most accurate (97%) for the identification of sex. This algorithm allows for quick and accurate testing of DNA markers and can greatly improve the efficiency of date palm genetics and cultivation due to the fact that the genotypes of male and female individuals can be identified before ripening, therefore reducing the costs and time of cultivation. Albarrak et al. ([63]) proposed a network model based on the following architects: AlexNet, VGG16, InceptionV3, ResNet, and MobileNetV2, of which MobileNetV2 was the most precise, with an accuracy of 97%. The authors verified the network on data from eight different classes of date palm fruits, and in order to increase the accuracy, they used various image processing techniques, e.g., magnification, model checkpoints, and hybrid adjustment.

Jintasuttisak et al. [70] used a CNN model and the YOLO-V5 architecture to detect date palms in images captured by a camera on a drone flying at an altitude of 122 m over agricultural fields. To prepare the data, they randomly selected 125 images and divided them into three datasets: training (60%), validation (20%), and testing (20%). The results of using the YOLO-V5 architecture to detect date palm in photos taken by the drone were compared by the authors with the results from other popular CNN architectures, i.e., YOLO-V3, YOLO-V4, and SSD300. The results show that the YOLO-V5m model has the highest accuracy, with an average precision of 92.34%. In addition, this neural network architecture enables the detection and localization of date palms of different sizes among different plants and occurring in different environments, in which date palms are relatively rare. The authors concluded that this method could be a useful part of an automatic date palm plantation management system and make it easier to forecast the volume of fruit production and monitor the condition of date palms. Muhammad [65] applied machine learning methods to classify date palm fruit types using local binary pattern (LBP) and Weber local descriptor (WLD) for textures combined with geometric features, dimensions, and shape. The author classified images of four types of dates, achieving a model accuracy of 98.1%.

Haidar et al. [64] achieved a classification accuracy of 98.6% for date palm fruits based on their species using neural networks and fifteen parameters, including colour, shape, geometric dimensions, and texture. However, it should be noted that only 140 images were used for date classification. Oussama and Kherfi [71] used a much larger dataset of 5000 images for the classification of 10 date palm fruit varieties. They used an RGB colour matrix, grayscale levels, and four shape features combined with a Gaussian Mixture Model (GMM), achieving a model accuracy of 97.5% with a classification time of 0.029 s. Similar to this study, images of a single date with a uniform background were used.

Zhang et al. [61] used image analysis and backpropagation colour decomposition to classify one date type into four ripeness classes in images of single fruits, achieving an accuracy of 97.5%. In their study, Pourdarbani et al. [51] presented a model for classifying dates into four ripeness classes through the taxonomic classification method with RGB colours and texture features, achieving an overall accuracy of 88.33% with a classification time of 0.34 s. In more recent publications, Hossain et al. [72] proposed date classification based on a pre-trained CNN model, achieving an accuracy of 99.2%. They used images of both single and multiple dates for four varieties. However, it is noticeable that the imaged fruits were characterised by high inter-species variability.

## 5. Conclusions

The paper proposes a CNN architecture and a model DateNET for automatic classification of date palm fruits of five varieties based on their colour and geometric parameters. The fixed CNN architecture consists of an alternating system of five Conv2D, MaxPooling2D, and Dropout classes, for which a computational algorithm was developed in the Python 3.9 language. The developed algorithm, described in code, allows the user to define any number of classes (varieties of date palm fruits) and analyse any number of images copied to training, validation, and test catalogues, which greatly increases its utility.

The validation accuracy of the model presented in this study depended on the selection of classification criteria. It was 85.24% for fruit colour-based classification and 87.62% for the geometric parameters only; however, it increased considerably to 93.41% when both the colour and geometry of dates were considered.

To sum up, it can be concluded that the classification of diverse fruits like dates solely based on geometric parameters, even within one variety, is a complicated process, as dates are characterised by great diversity in terms of texture, and the DateNET model can treat them as distinct.

In the DateNet model, the ability to recognise geometric fruit parameters is the main advantage over the models for which the only classification criterium is colour, as it allows for stimultaneuos distinction. With appropriate technical equipment, the proposed model can provide measurable benefits, despite the longer classification time compared to the existing ones.

## Figures and Tables

**Figure 1 sensors-24-00558-f001:**
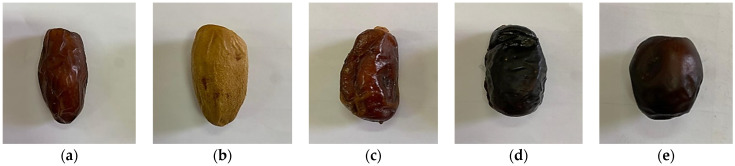
Imaged date palm fruit at the Tamr stage of varieties: (**a**) Deglet Nour, (**b**) Degla Baida, (**c**) Ghars, (**d**) Tenicin, (**e**) Tantboucht.

**Figure 2 sensors-24-00558-f002:**
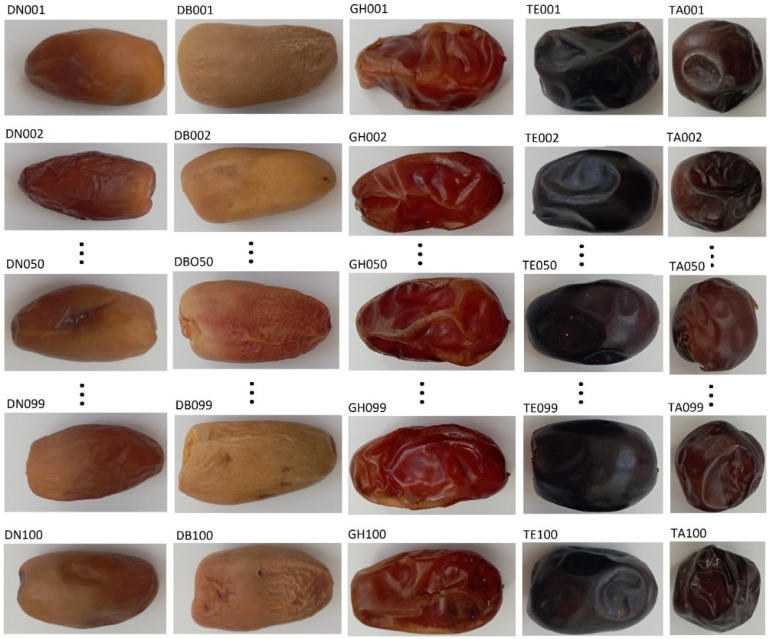
Visualisation of date palm fruit images.

**Figure 3 sensors-24-00558-f003:**
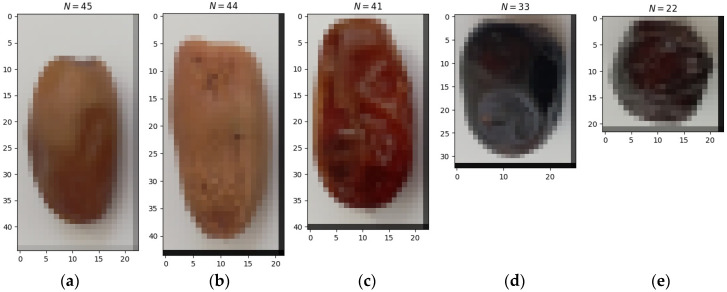
Discretized images of date palm fruit varieties: (**a**) Deglet Nour, (**b**) Degla Baida, (**c**) Ghars, (**d**) Tenicin, (**e**) Tantboucht.

**Figure 4 sensors-24-00558-f004:**
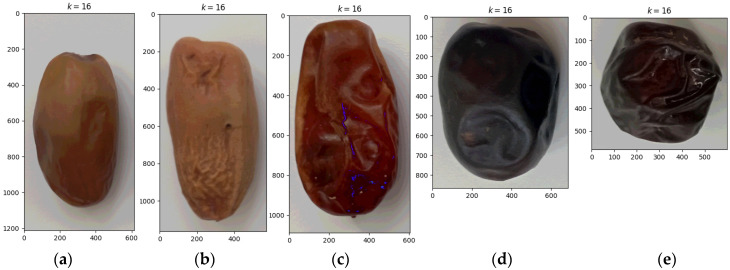
Quantified images of date palm fruits of varieties: (**a**) Deglet Nour, (**b**) Degla Baida, (**c**) Ghars, (**d**) Tenicin, (**e**) Tantboucht.

**Figure 5 sensors-24-00558-f005:**
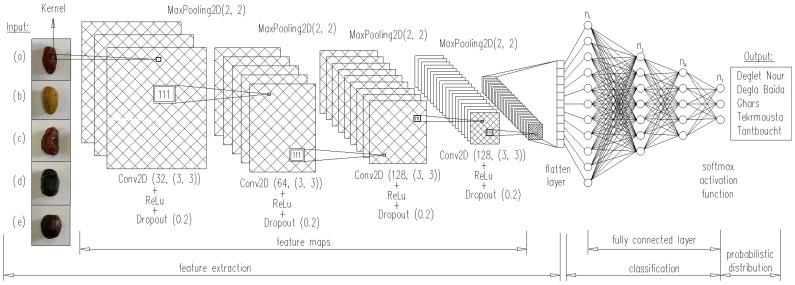
Schematic of the implemented CNN network (DateNET). (**a**) Deglet Nour, (**b**) Degla Baida, (**c**) Ghars, (**d**) Tenicin, (**e**) Tantboucht.

**Figure 6 sensors-24-00558-f006:**
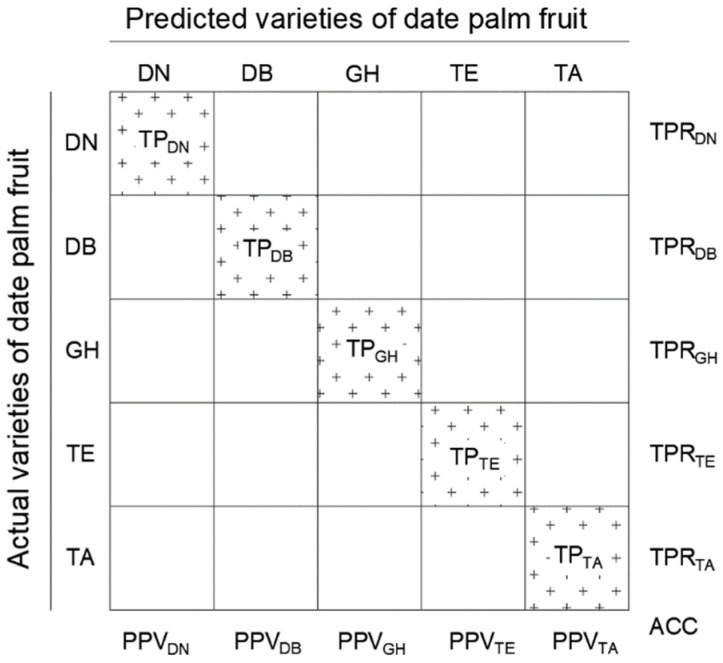
Confusion matrix scheme for the classification of date palm fruit varieties. “+” is the background/fill.

**Figure 7 sensors-24-00558-f007:**
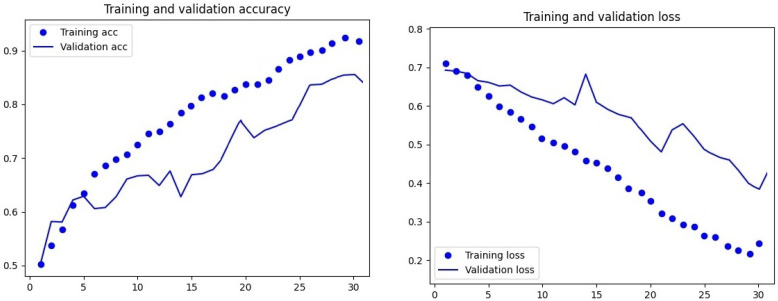
Visualisation of loss function curves and learning accuracy and model validation, based on date palm fruit colour, for 30 cycles of computational repetition of the created DateNET.

**Figure 8 sensors-24-00558-f008:**
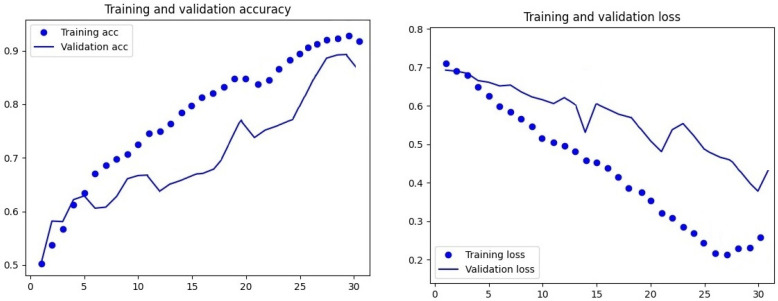
Visualisation of loss function curves and learning accuracy and model validation, based on colour and geometric parameters of date palm fruit, for 30 cycles of computational repetitions of the created DateNET.

**Figure 9 sensors-24-00558-f009:**
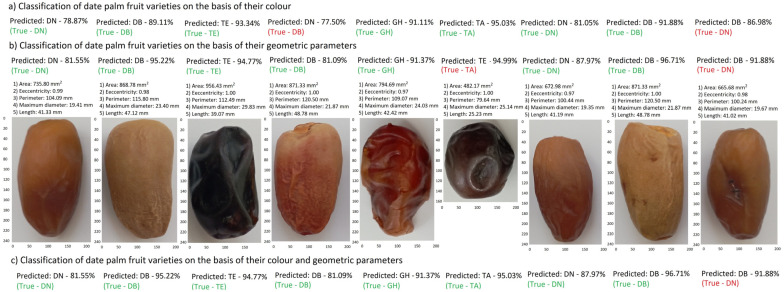
Example outputs of imaged date palm fruits with their predicted labels. red color is a misclassification.

**Figure 10 sensors-24-00558-f010:**
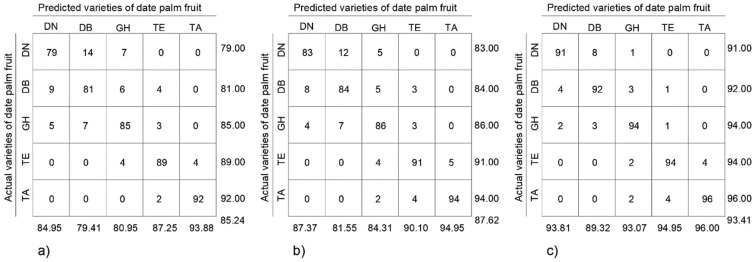
Confusion matrix of date palm fruit variety classification model DateNET based on: (**a**) colour, (**b**) geometric parameters, (**c**) colour and geometric parameters; DN-Deglet Nour, DB-Degla Baida, GH-Ghars, TE-Tenicin, TA-Tantboucht.

**Table 1 sensors-24-00558-t001:** Morphological characterization of dates varieties.

Fruit (Tamr Stage)	Deglet Nour	Degla Baida	Ghars	Tenicin	Tantboucht
Fruit’ shape	Sub cylindrical	Sub cylindrical	Sub cylindrical	Sub cylindrical	Spherical
Fruit shape at distal end	Flat sharp	Flat sharp	Oval oblique	Oval oblique	Large round
Fruit shape of stalk end	Oval oblique	Oval oblique	Sharp	Flat oval	Large round
Average fruit length (mm)	41.20 ± 0.10	47.20 ± 0.11	42.20 ± 0.16	39.60 ± 0.14	25.80 ± 0.08
Average fruit diameter (mm)	19.20 ± 0.53	22.62 ± 0.11	22.20 ± 0.16	29.60 ± 0.14	25.89 ± 0.08
Average fruit weight (g)	9.39 ± 0.17	9.92 ± 0.44	8.18 ± 0.61	7.45 ± 0.79	8.91 ± 0.62
Colour at Tamar stage	Brown	Yellowish brown	Brown	Blackish brown	Blackish brown
Fruit consistency	Semi-Dry	Dry	Soft	Semi-Soft	Semi-Soft
Aspect of fruit epicarp (skin)	Semi-Smooth	Semi-Smooth	Smooth	Smooth	Smooth
Alteration of the epicarp (skin) colour	Brown	Yellowish brown	Dark brown	Black	Black
Fruit flesh texture	Fibrous	Mealy	Fibrous	Fibrous	Fibrous
Fruit flesh colour	Whitish yellow	White	Yellow	Whitish yellow	Whitish yellow
Taste and flavor of the date	Sugary	Sugary	Sugary-acidulous	Sugary-acidulous	Sugary-acidulous
Fruit aroma	Present	Absent	Present	Present	Present
Length of the date cavity (mm)	35.20 ± 0.08	39.60 ± 0.10	37.80 ± 0.09	32.60 ± 0.10	20.90 ± 0.09
Width of the date cavity (mm)	09.40 ± 0.05	11.50 ± 0.04	09.30 ± 0.04	09.20 ± 0.04	15.00 ± 0.08
Length of the seed (mm)	2.65 ± 0.02	3.02 ± 0.09	2.70 ± 0.03	2.59 ± 0.10	1.88 ± 0.037
Average seed weight (g)	0.83 ± 0.04	1.75 ± 0.11	0.96 ± 0.11	0.86 ± 0.13	0.99 ± 0.05
Seed shape	Sub cylindrical	Sub cylindrical	Sub cylindrical	Sub cylindrical	Oval
Harvest period	Late (In November)	Late (In November)	Early (At the end of August)	Early (At the end of August)	Early (At the end of August)
Utilization (by-product)	Export for all uses	Flour	Used as a paste and Robe	Used as Robe	Used as a paste and Robe

**Table 3 sensors-24-00558-t003:** Criteria for the geometric classification of date palm fruit.

Fruit (Tamr Stage)	Deglet Nour	Degla Baida	Ghars	Tenicin	Tantboucht
Average fruit length (mm)	41.20 ± 0.10	47.20 ± 0.11	42.20 ± 0.16	39.60 ± 0.14	25.80 ± 0.08
Minimum fruit length (mm)	41.10	47.09	42.04	39.46	25.72
Maximum fruit length (mm)	41.30	47.31	32.36	39.74	25.88
Average fruit diameter (mm)	19.20 ± 0.53	22.62 ± 0.11	22.20 ± 0.16	29.61 ± 0.14	25.89 ± 0.08
Minimum fruit diameter (mm)	18.67	22.51	22.04	29.47	25.81
Maximum fruit diameter (mm)	19.73	22.73	22.36	29.75	25.97

Source: own study based on empirical research.

**Table 4 sensors-24-00558-t004:** Changing the size of DateNET model maps depending on layer number.

Layer (Type)	Output Shape	Param
conv2d (Conv2D)	(None, 198, 198, 32)	645
max_pooling2d (MaxPooling2D)	(None, 99, 99, 32)	0
dropout (Dropout)	(None, 99, 99, 32)	0
conv2d_1 (Conv2D)	(None, 97, 97, 64)	12,421
max_pooling2d_1 (MaxPooling2D)	(None, 48, 48, 64)	0
dropout_1 (Dropout)	(None, 48, 48, 64)	0
conv2d_2 (Conv2D)	(None, 46, 46, 128)	59,959
max_pooling2d_2 (MaxPooling2D)	(None, 23, 23, 128)	0
dropout_2 (Dropout)	(None, 23, 23, 128)	0
conv2d_3 (Conv2D)	(None, 21, 21, 128)	121,664
max_pooling2d_3 (MaxPooling2D)	(None, 10, 10, 128)	0
dropout_3 (Dropout)	(None, 10, 10, 128)	0
flatten (Flatten)	(None, 12,800)	0
dense (Dense)	(None, 512)	4,454,424
dense_1 (Dense)	(None, 1)	527

Total params: 4,459,332; trainable params: 4,459,332; non-trainable params: 0.

**Table 5 sensors-24-00558-t005:** Geometric parameters of dates from labels generated by the DateNET model.

Code Photos	Area [mm^2^]	EC	PE [mm]	MD [mm]	DAD [mm]	Length [mm]	DAL [mm]	DFV	ΔE	CR
1	2	3	4	5	6	7	8	9	10	11
DN001	735.80	0.99	104.09	19.31	+0.11	41.03	+0.03	DN	2.7	True
DN002	672.98	0.97	100.44	19.25	+0.05	41.19	−0.01	DN	2.8	True
DN003	665.68	0.98	100.24	19.27	+0.07	41.12	−0.08	DN	2.6	True
DN004	554.74	0.99	93.74	18.98	−0.22	41.30	+0.10	DN	2.4	True
DN005	663.45	1.00	98.75	19.25	+0.85	40.99	−0.21	DN	2.9	True
…	…	…	…	…	…	…	…	…	…	…
DN099	635.77	0.98	101.22	19.25	+0.05	41.04	−0.16	DN	2.0	True
DN100	555.65	0.99	100.24	19.17	−0.03	41.12	−0.08	DN	2.7	True
DB001	764.69	0.99	109.63	21.84	−0.78	46.68	−0,52	DB	3.6	True
DB002	871.33	1.00	120.50	21.87	−0.75	48.78	+1.58	DB	2.9	True
DB003	853.85	0.99	116.33	22.89	+0.27	46.83	−0.37	DB	1.9	True
DB004	924.10	0.95	123.20	23.10	+0.48	49.55	+2.35	DB	2.4	True
DB005	868.78	0.98	115.80	23.40	+0.78	47.12	−0.08	DB	2.2	True
…	…	…	…	…	…	…	…	…	…	…
DB099	851.85	0.99	117.37	22.87	+0.25	45.83	−1.37	DB	3.2	True
DB100	921.10	0.97	125.20	23.15	+0.53	47.55	+0.35	DB	0.7	False
GH001	740.18	0.99	105.25	21.65	−0.55	40.88	−1.32	GH	4.2	True
GH002	764.18	0.99	108.78	22.09	−0,11	43.36	+1.16	GH	4.5	True
GH003	794.69	0.97	109.07	24.03	+1.83	42.42	+0.22	GH	5.1	True
GH004	761.43	0.95	109.73	23.15	+0.95	41.66	−0.54	GH	4.9	True
GH005	701.84	1.00	106.41	20.06	−2.14	40.83	−1.37	GH	5.0	True
…	…	…	…	…	…	…	…	…	…	…
GH099	762.43	0.95	108.78	22.15	−0.05	41.68	−0.52	GH	2.9	True
GH100	721.84	1.00	107.47	21.06	−1.14	41.23	−0.97	GH	3.1	True
TE001	956.43	1.00	112.49	29.83	+0.22	39.07	−0.53	TE	5.9	True
TE002	982.99	0.99	115.16	31.62	+2.01	39.77	+0.17	TE	3.6	True
TE003	805.70	1.00	104.40	29.16	−0.45	38.90	−0.70	TE	3.3	True
TE004	817.35	0.98	196.87	28.83	−0.78	39.12	−0.48	TE	0.9	False
TE005	873.47	0.99	108.70	28.57	−1.04	40.02	+0.42	TE	3.9	True
…	…	…	…	…	…	…	…	…	…	…
TE099	805.75	1.00	105.45	29.15	−0.46	38,90	−0.70	TE	4.4	True
TE100	815.55	0.98	195.85	28.85	−0.76	39.12	−0.48	TE	5.5	True
TA001	482.17	1.00	79.64	25.14	−0.75	25.23	−0.57	TA	4.2	True
TA002	482.09	1.00	79.00	24.67	−1.22	25.45	−0.35	TA	5.4	True
TA003	490.13	1.00	80.17	24.92	−0.97	24.48	−1.32	TA	4.7	True
TA004	499.18	1.00	80.78	25.18	−0.71	25.04	−0.76	TA	3.6	True
TA005	612.14	1.00	91.73	29.52	+3.63	24.44	−1.36	TA	4.1	True
…	…	…	…	…	…	…	…	…	…	…
TA099	485.09	1.00	79.55	24.56	−1.33	25.55	−0.25	TA	5.6	True
TA100	490.44	1.00	80.44	24.94	−0.95	24.44	−1.36	T TA	1.2	False

EC—eccentricity, PE—perimeter, MD—maximum diameter, DAD—deviation from average diameter, DAL—deviation from average length, DFV—date palm fruit variety, CR—correction.

**Table 6 sensors-24-00558-t006:** Performance of the proposed date palm fruit variety classification model DateNET.

Classification Type	ACC [%]	PPV [%]	TPR [%]	fscore [%]	Average Classification Time GPU * [ms/Image]
Colour	85.24	85.29	85.20	86.89	20.43
Geometric parameters	87.62	87.66	87.60	90.76	23.67
Colour and geometric parameters	93.41	93.40	93.40	95.03	27.88

* GPU: NVIDIA GeForce RTX Studio 2060, 32 GB (NVIDIA, Santa Clara, CA, USA).

**Table 7 sensors-24-00558-t007:** Comparison of DateNet, ResNet, Mobilenet, and ShuffleNet classification performance and time.

Classification Type	ACC [%]	PPV [%]	TPR [%]	fscore [%]	Average Classification Time GPU * [ms/Image]
DateNET	93.43	93.41	93.41	95.03	27.89
ResNet	93.24	94.27	94.28	94.09	20.77
MobileNet	91.98	94.27	94.27	94.27	21.42
ShuffleNet	89.62	91.66	91.66	90.76	22.66

* GPU: NVIDIA GeForce RTX Studio 2060, 32 GB.

## Data Availability

Data are contained within the article.

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
