# Peer review of "Convolutional Neural Network (CNN) Model for the Classification of Varieties of Date Palm Fruits (Phoenix dactylifera L.)"

_sensors, 2024, doi:10.3390/s24020558_

Round 1

Reviewer 1 Report

Comments and Suggestions for Authors

In the presented paper authors tackle the issue of classification of dates by their variety. It is an interesting study and authors seem to achieve some satisfactory results; however, I believe that method’s description could be improved. Some comments I have in particular are listed below.

1.     In Section 2.5 authors mention «the proposed algorithm» multiple times; however, I couldn’t locate the clear algorithm description in the paper. Could the authors clarify what they mean by the algorithm? Some particular questions to the algorithm:

1.1.  Looking at Figure 5, where do image features such as colour and geometry come into play here?

1.2.  In line 290:Could authors clarify what they mean by binary classification here? As far as I am aware, the classification into two classes is called binary; but it seems that there are more classes in the model.

1.3.  Line 294: How does the algorithm performs the sorting? Does algorithm evaluates geometric parameters of dates or this data is pre-measured manually?

1.4.  Line 296: What is meant by «the appropriate directory» here?

1.5.  Line 380: In which proportions the sorting was performed?

I think the paper would benefit from a clear description of the steps involved in the algorithm in chronological order.

Some other concerns:

2.     Section 2.1. Where does the data in Table 1 come from? Is it collected by authors or taken from some source?

3.     Section 2.4. The phrase «In practice, images are characterized by a large excess of information …» repeats twice.

4.     Section 2.4. Table 3. Was the study for the table conducted on the same dataset that was used for testing?

5.     Section 2.5. Figure 5. It seems that legibility of the text could be improved.

6.     Section 2.5. Equations (5) and (6) seem to be identical.

7.     Lines 381-383: The same phrase is repeated.

Overall, I believe that the paper could by published after some revisions.

Author Response

Dear Reviewer 

The answers to your questions and comments are contained in the submitted file.

Reviewer 2 Report

Comments and Suggestions for Authors

The paper proposes a CNN architecture and a model DateNET for automatic classification of date palm fruits of five varieties, based on their colour and geometric parameters. The fixed CNN architecture consists of an alternating system of five Conv2D, MaxPooling2D and Dropout classes for which a computational algorithm was developed in the Python 3.9 language. The developed algorithm, described in code, allows the user to define any number of classes (varieties of date palm fruits) and analyse any number of images copied to training, validation and test catalogues, which greatly increases its utility. It is recommended that the author consider image samples of date palm fruits under different growth cycles and meteorological conditions when establishing a recognition model, in order to enhance the robustness of the model.

Comments on the Quality of English Language

The quality of English writing is acceptable.

Author Response

Dear Reviewer 

The answers to your questions and comments are contained in the submitted file 

Reviewer 3 Report

Comments and Suggestions for Authors

This paper implements an image-based method to evaluate date quality
rather than using a CNN model for contact detection, and performs
experiments on it. The article is well written but lacks innovation. I
have some questions and suggestions for this article, which are listed
below.

1. The CNN in the innovative method is unclear and the model diagram is
unreasonable. It is recommended to read more relevant literature and
learn how to draw model diagrams.

2. The purpose of this study is to achieve non-contact classification,
aiming to solve the quality of dates caused by contact grading. At the
same time, it is not proposed that if there is deep learning evaluation
method used or not. It is recommended to compare it with traditional
contact-based methods in comparative experiments rather than comparing
accuracy with other models

3. When comparing with other models, ablation experiments should also be
performed. What changes have been made to this model and the impact of
these changes needs to be demonstrated.

4. This article compares the accuracy performance of multiple models in
the red date classification task. Among them, ResNet has a very close
accuracy to the model proposed in this article. At the same time, the
conclusion shows that the model proposed in this paper takes longer, and
the advantages of the model are reflected.

Author Response

(The authors gave the same response as above.)

Round 2

Reviewer 1 Report

Comments and Suggestions for Authors

The authors have considered the reviewers comments and made some appropriate changes. I have no other comments.

Reviewer 3 Report

Comments and Suggestions for Authors

Accepted